# A Bayesian framework to unravel food, groundwater, and climate linkages: A case study from Louisiana

**Nitin K. Singh** [1,2]*, **Ruchi Bhattacharya**[1], **David M. Borrok**[2]

**1** Earth and Environmental Sciences, University of Waterloo, Waterloo, Ontario, Canada, **2** Geosciences and Geological and Petroleum Engineering, Missouri University of Science and Technology, Rolla, MO, United States of America

* nksingh2@ncsu.edu

**Data Availability Statement:** The datasets used in the study are publicly available from NOAA, NCDC, NASS, USGS, and Louisiana DNR. Citations are available within the manuscript.

## Abstract

Advancing our understanding of the connections among groundwater, food, and climate is critical to meet global food demands while optimizing water resources usage. However, our understanding of the linkages among groundwater, food, and climate is still limited. Here, we offer a Bayesian framework to simulate crop yield at a regional scale and quantify its relationships and associated uncertainty with climate, groundwater, agricultural, and energy-related variables. We implemented the framework in the rice-producing regions of Louisiana from 1960–2015. To build a parsimonious model, we used a probability-based variable selection approach to detect the key drivers of rice yield. Rice yield increased, groundwater declined, and area planted declined or did not change over 56yrs. The number of irrigation wells, groundwater level, air temperature, and area planted were found to be the key drivers of rice yield. The regression coefficients showed that rice yield was positively related to groundwater level, and negatively related to area planted and the number of irrigation wells. The limited influence of N fertilizer was noted on rice yield for the period when fertilizer data were available. The inverse relationship between rice yield and area planted pointed to the adaption of efficient crop management practices that maintained or increased yield, despite the decline in area planted. The farmers' ability to install irrigation wells during droughts sustained the yields over long-term but not short-term. This decline in rice yield in response to drought over the short-term might explain the negative relation between yield and irrigation wells. Overall, this work highlighted the uncertainty in relationships between rice yield and key drivers and quantified the intimate connection between food and groundwater. This work may have implications for managing two highly competing commodities (i.e., groundwater and food) in agricultural regions.

## 1. Introduction

In 2011, the Food and Agriculture Organization projected that due to the rising population, global demand for food and freshwater is expected to increase more than 60% by 2050 [1,2].

**Funding:** DMB: National Science Foundation (CBET 1360398) https://www.nsf.gov.

**Competing interests:** The authors have declared that no competing interests exist.

Due to the combination of unsustainable irrigation and drought [3], groundwater continues to decline at an alarming rate globally [4]. Despite overall increases in groundwater used for irrigation, crop yields in some regions of the world have stagnated or declined [5]. For instance, 37% of the rice acreage at the global scale exhibited a decline or no change in rice yield from 1961 to 2008 [5]. Such patterns may have severe implications for food security in the near future. Given the importance of food security and freshwater availability to society, there is an urgent need to advance our understanding of the intimate linkages between crop production and groundwater level. In turn, this may lead to the sustainable management of groundwater resources while meeting global food demands [6].

Several process-based crop growth models have been proposed to simulate yield based on climatic, agricultural, landscape, and physiological variables. The process-based crop growth models broadly simulate mechanisms that account for plant development, soil conditions, and water management practices from plot to regional scales [7–16]. Such detailed consideration of processes requires several variables to simulate crop yield, resulting in high uncertainty in the predictions [17]. Further, the lack of availability of a range of datasets needed to build a process-based model becomes challenging for data-scarce regions with limited resources [18]. These issues have led to the development of statistical models to simulate crop yields [17,19].

Generally, statistical models used regression-based approaches to simulate crop yield from regional to global scales [20–28]. For instance, a linear regression model was used to investigate relationships of corn and soybean yields with climatic variables, such as precipitation and air temperature at the county scale across the United States [21]. Similarly, regression-based models were used to quantify the linkages between rice yield and climatic drivers such as radiation, temperature, and precipitation in China [24] and Philippines [21]. In a seminal study, multiple linear regression equations were used to attribute the spatiotemporal patterns of six crop types to climatic drivers on a global scale [25]. These studies collectively demonstrate the utility of regression-based models in revealing the controls of crop yield across spatial scales.

Largely, the past statistical crop yield models have been limited in their scope in two distinct ways. First, these statistical studies were focused on quantifying the effect of climate change on crop yield, so the inclusion of groundwater, agricultural, or energy-related datasets in models have been rare. For instance, the direct linkages between groundwater and crop production have been documented across many agricultural regions [28–34]. Using scenario-based statistical analysis, the authors showed how declining groundwater might influence corn production in the near future [29]. A modeling study from the North China Plains demonstrated that limiting groundwater irrigation can lead to 40% reduction in crop production [32]. Recently, causal linkages between groundwater levels and rice yield have been estimated over 50 years in the agricultural regions of Louisiana [35], where irrigation is mostly dominated by pumping [36]. At the same time, energy-related variables have been shown to influence the production of agricultural commodities [37–38]. For instance, patterns of wheat yield have been attributed to energy inputs such as energy fuels, electricity [39]. However, energy variables are rarely considered in the crop yield models. Thus, the linkages among food, energy, and water are crucial for society but remain understudied in this context [40]. Second, most of the past statistical models relied on deterministic relationships to simulate crop yield, and the limited attempts have been made to explore the uncertainty in relationships between crop yield and associated explanatory variables [41]. Generally, climatic and environmental drivers are highly heterogeneous and vary widely in space and time. Groundwater and climatic variables are expected to change due to the rise in population and climate change, and available datasets may not be adequate to reflect all possible combinations of outcomes. Thus, incorporating uncertainty is critical for making informed decisions and advancing our understanding of food, climate, and water nexus in the near future. Therefore, there is a need to develop an approach that can

simulate crop yields while systematically exploring uncertainty and including critical food, energy, and groundwater variables.

Here, we develop a robust yet simple Bayesian inference based framework that simulates crop yield with commonly available agricultural, climatic, energy, and groundwater-related datasets, while incorporating uncertainty in the model parameters. To build a parsimonious Bayesian model, we implemented a variable selection approach to determine the most important controls of crop yield. We tested this framework at a regional scale in the rice-producing region of Louisiana, where groundwater is under stress due to intensive irrigation [36, 42]. We assembled a combination of climatic, groundwater, energy, and agricultural datasets from several publicly available databases such as National Centers of Environmental Information (NCEI), the National Agricultural Statistics Service (NASS), United States Geological Survey (USGS), and the Louisiana Department of Natural Resources (LDNR) from 1960 to 2015. The objectives of the study were to: (i) explore and simulate the spatiotemporal patterns of rice yield, and (ii) to quantify its linkages with key factors including climate (e.g., rainfall totals, air temperature), groundwater levels, agriculture (e.g., area planted, number of irrigation wells, fertilizers) and energy (e.g., oil prices). The study was conducted in the rice-producing regions of Louisiana. However, the proposed framework can be extended to other agricultural regions of the world where the datasets used in the study are generally available or could be estimated.

## 2. Materials and methods

### 2.1 Study sites

The study was conducted in the rice-producing counties of Louisiana that had the necessary long-term (>50yrs) data for the range of time series used in the study (S1 Fig). The counties that were considered in the study include Acadia (AC), Beauregard (BE), Cameron (CN), Evangeline (EV), East Carroll (EC), Jefferson Davis (JD), Iberia (IB), St. Martin (SM), Vermillion (VE), and West Carroll (WC).

### 2.2 Datasets

Table 1 summarizes details regarding the datasets used in the study. The annual time series of rice yield, the area planted, and the total number of wells installed for irrigation at the county level from 1960 to 2015 were obtained from the National Agricultural Statistics Service (NASS) and Louisiana Department of Natural Resources. Crude oil prices have been shown to influence the production and prices of agricultural commodities [37–38]. For example, a study recently demonstrated a strong relationship between food prices and energy prices over the past decade [43]. Therefore, we used crude oil prices as a surrogate for energy in our work. The price of crude oil was adjusted for inflation to 2015 prices.

Groundwater levels (depth from the land surface) were obtained from the USGS groundwater database. A well with the most available data within each county at the annual timescales from 1960–2015 was selected for the analysis. The wells are located in the Chicot aquifer, which is part of the larger Coastal Lowland Aquifer system (e.g., AC, BE, CN, EV, JD, IB, SM, VE), and in the Mississippi River Valley Alluvial aquifer system (e.g., EC, WC) [50,51]. Spanning over 23000 km$^2$, the Chicot aquifer is comprised of sequence of clays, gravel, sand, and silt constitutes at varying depths [50]. The aquifer thickness can be as great as 700 feet at places in Louisiana [50]. The Chicot aquifer is the major sources of fresh groundwater for the region, where the majority (70%) of the freshwater is withdrawn for irrigation purposes [52]. The Lower Mississippi River Valley Alluvial aquifer system mostly consists of unconsolidated sands that are interbedded and frequently capped by silt and clay. The aquifer thickness can

**Table 1. Summary of variables used in the study.**

| Variables | Description | Source | Spatial Scale of data availability |
|---|---|---|---|
| Seasonal Rainfall totals (mm) | Daily rainfall depths were aggregated over growing seasons | NCEI [44] | North gauging station for (EC, WC counties) and South gauging station (AC, BE, CN, EV, JD, IB, SM, VE counties) |
| Mean Air Temperature (Tmean,°) | Daily mean air temperatures were averaged over growing season | NCEI [44] | North gauging station for (EC, WC counties) and South gauging station (AC, BE, CN, EV, JD, IB, SM, VE counties) |
| Palmer Drought Severity Index (PDSI) | Proxy for antecedent conditions [45]; Monthly PDSI values were averaged for the growing season | NCEI [44] | County Scale |
| Rice Yield (lb/acre) | Total rice produced per unit area at annual scale | NASS [46] | County Scale |
| Area Planted (ha) | Total area of rice planted at annual scale | NASS [46] | County Scale |
| Fertilizer Inputs (TN/TP) | Fertilizer totals | [47] | County Scale |
| Number of Irrigation wells | Total number of wells installed annually for irrigation | LDNR [48] | County Scale |
| Groundwater Level (m) | Mean groundwater level from the surface | USGS [49] | County Scale |
| Oil price (USD) | Nominal crude oil price was adjusted for inflation to 2015 prices | | US Scale |
| Annual Rainfall totals (mm) | Daily rainfall depths were aggregated at annual scale | NCEI [44] | North gauging station for (EC, WC counties) and South gauging station (AC, BE, CN, EV, JD, IB, SM, VE counties) |

range from 25 feet to 150 feet [51]. The Lower Mississippi River Valley Alluvial aquifer is the one of the heavily used aquifer in the United States [53].

Two representative climate stations (one in the north for EC, WC, and one in the south for AC, BE, CN, EC, EV, IB, JD, SM) were used to retrieve mean air temperature and rainfall totals from 1960 to 2015. The rainfall totals and mean air temperature were computed at the growing season scale over the tested 56 years (S2 Fig). Studies have also shown the influence of annual rainfall totals on rice yield [54], which led us to consider it as a potential covariate for the model. The Palmer Drought Severity Index (PDSI) has been extensively used in understanding antecedent conditions [45,55]. The PDSI accounts for soil moisture-holding capacity and evapotranspiration via physical water balance models [56]. For our study, monthly PDSI values at the county level from 1960 to 2015 were obtained from the Climate Data Online of National Centers of Environmental Information (NCEI). Further, monthly PDSI values were aggregated for growing seasons at the county level during the study period. The growing season for the rice in this region is February through July. The total nitrogen and total phosphorus fertilizer inputs for the study counties were available at an annual scale from 1987–2012 [47]. We normalized the fertilizer inputs (tons) with the rice area planted within each county.

## 2.3 Statistical modeling

In order to build a parsimonious model, we implemented a probability-based variable selection approach to determine the importance of explanatory variables (X) for rice yield (Y) [57]. This approach was also critical in addressing the issue of collinearity, which has been highlighted in the regression-based crop models [17]. Initially, we built models to exhaust all possible combinations of variables ($2^K$; K = number of variables). Later, model ensembles were used to find the probability of inclusion of each variable, depending upon their explanatory power when they were included in the model. In other words, if the probability of inclusion for a variable was about 1, it means the variable had the greatest explanatory power when it was included in the model. The variable selection approach was conducted using the R 2.5.1 software [58].

We developed hierarchical Bayesian regression models to simulate rice yield and explore the posterior distributions of regression coefficients for the key potential drivers derived from the variable selection approach. The Bayesian estimation approach allowed us to incorporate the uncertainty in relationships between rice yield and the drivers. Owing to missing data and limited observations (~20) for some counties, we integrated all observations and developed a fully pooled hierarchical Bayesian regression models for the entire rice producing region.

The multilevel Bayesian regression model included data (Eq 1) and process (Eq 2) models. All drivers and the response variable were of different magnitude and scale, so for an unbiased comparison of regression coefficients among drivers, the response variable and all drivers were standardized before fitting the model [59]. In addition to the major drivers that may influence rice yield, we used time as a factor in the model, as suggested [60].

$$\boldsymbol{\theta}|y \sim N(\mu, \tau) \qquad \text{Eq (1)}$$

$$\mu = a + \beta_1 * X_1 + \beta_2 * X_2 + \beta_3 * X_3 + \beta_4 * X_4 \ldots \beta_n * X_n \qquad \text{Eq (2)}$$

where $\boldsymbol{\theta}$ represents the distribution of all unknown parameters, y is observed rice yield, $\tau$ is precision (inverse of standard deviation) and $\mu$ is the mean of the projected distribution of rice yield, alpha is the intercept, and $\beta_1$-$\beta_n$ are the coefficients of the most important variables ($X_1$ to $X_n$). For simplicity, we refer to this long-term model with key covariates as 'model 1'. To test the role of fertilizers, we built another model with fertilizers plus the key drivers (Eq 2) for a limited duration when fertilizer datasets were available. Here onward, we refer to this limited duration fertilizer model as 'model 2'.

As a standard approach, we wanted data to inform our inference, so uninformative priors with uniform distributions were used for the parameters in data and process models (Eqs 1 and 2), and the gamma distribution based uninformative prior was used for the precision ($\tau$; Eq 1) [61]. Just Another Gibbs Sampler (JAGS) based on Gibbs sampling, a Markov Chain Monte Carlo algorithm, was used to estimate distributions of parameters in the R-JAGS [62] in R 2.5.1 [58]. We built four chains and ran 50000 simulations to assure the model convergence (i.e., $R_{hat}$<1.1) for all parameters [63]. The initial 40,000 simulations were discarded prior to parameter estimation. The Deviance Information Criteria (DIC) and classical coefficient of determination ($R^2$) were computed to assess the model fit. As part of the post predictive model check, we generated a new set of observations (i.e.,$y_{pred}$) at every iteration and compared them with actual observations of crop yield (y) as recommended [61]. If the model performed well, predicted values (i.e.,$y_{pred}$) should closely relate to the observed values (y).

## 3. Results

Rice yield showed a gradual increase over time, and the rate of increase was relatively steep during the last 30 years of the study period (Fig 1). On the contrary, groundwater level declined up-to 7m in the study counties (Fig 1), with a few exceptions where groundwater levels were highly variable (i.e., EC, JD) or changed minimally (i.e., IB). The temporal patterns of area planted showed mixed patterns (i.e., decrease, or no change) among counties during the study period (Fig 2). The Palmer Drought Severity Index (PDSI), a surrogate for antecedent conditions, exhibited a large number of negative PDSI values. The frequency of negative values was consistently higher in more recent years. We found that the high frequency of dry conditions corresponded to increases in the number of irrigation wells installed for most counties (Fig 3), indicating number of irrigation wells can also serve as a substitute for dry conditions. For the first three decades of the study period, the number of wells installed per county was

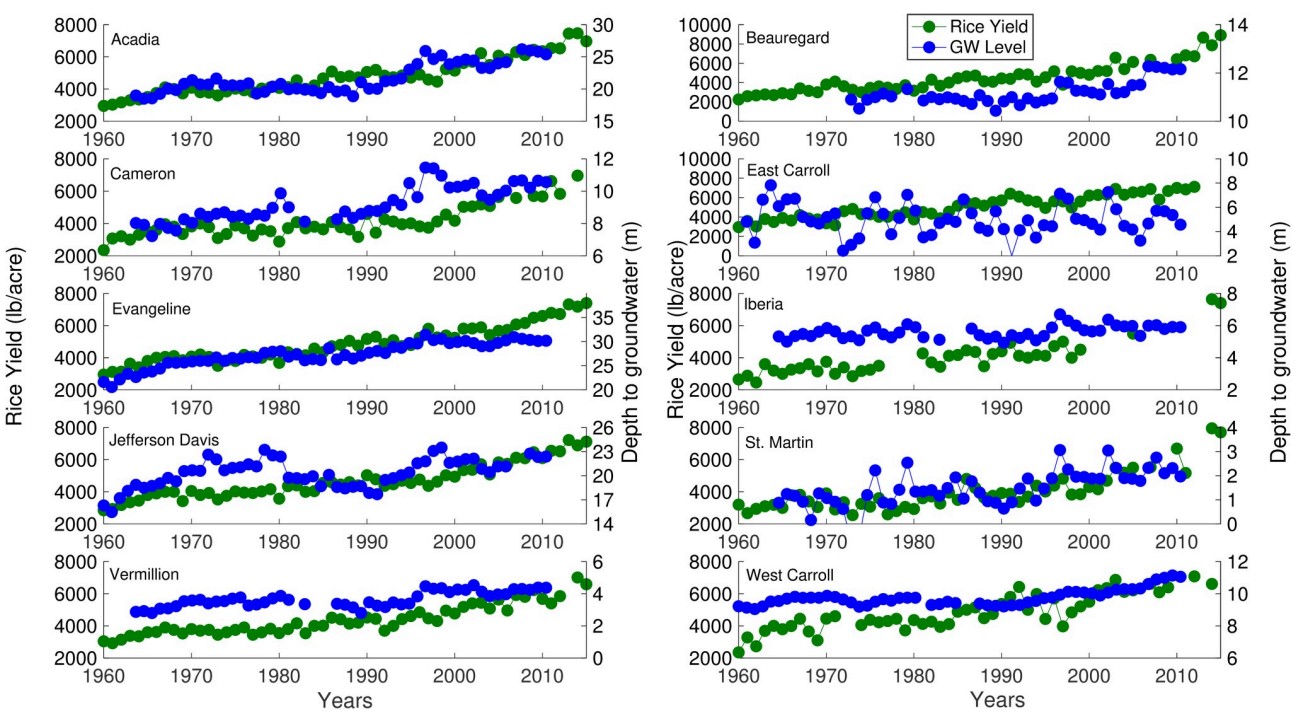

**Fig 1. Spatiotemporal patterns of rice yield and groundwater level across 10 counties in the state of Louisiana from 1960 to 2015.** The groundwater level is measured from the land surface, so greater the level drier the well.

less than 10. However, the number of irrigation wells dramatically increased near the end of the study period.

We found that cropped area normalized fertilizer inputs (N & P) did not show any consistent, unidirectional patterns for the limited years of data available (Fig 4). Further, a high correlation (r>0.85) was noted between N and P fertilizers. Due to the similarity in temporal patterns between N and P, and N being the commonly used fertilizer for rice production [64–65], we built model 2 using N fertilizer data. Lastly, oil prices from 1960 to 2015 varied widely with no clear temporal pattern (S3 Fig). S1 Table summarizes spearman's correlation coefficients among explanatory variables. A strong correlation (r>0.5) was only noted between growing season rainfall and PDSI and annual and seasonal rainfall totals (S1 Table). Thus, minimal correlations were noted among most of the explanatory variables.

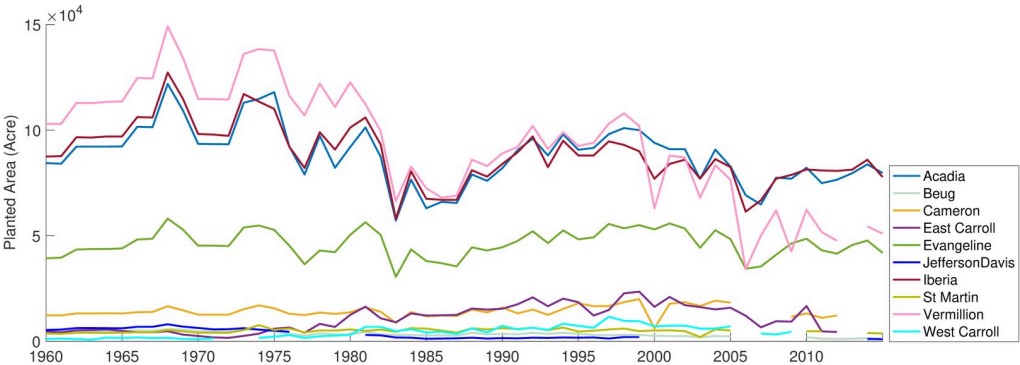

**Fig 2. Rice area planted during the 56 years across the study counties in Louisiana.**

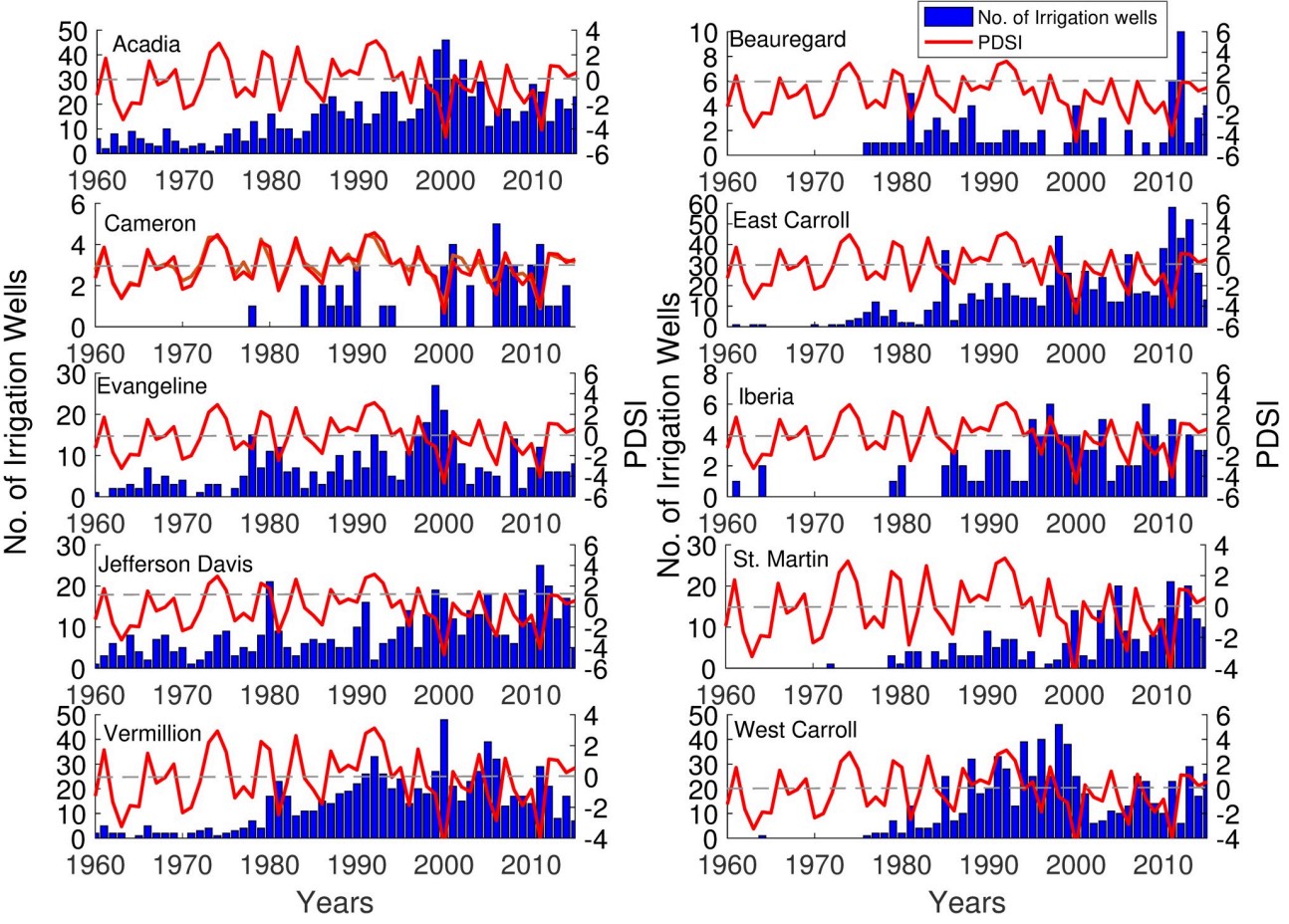

**Fig 3. Spatiotemporal patterns of Palmer Drought Severity Index (PDSI) and numbers of irrigation wells installed across 10 counties of Louisiana.**

A variable selection approach ranked the key drivers of rice yield for the study period (Table 2). The probability of inclusion was high (>0.9) for the number of irrigation wells, groundwater level, mean air temperature, and area planted, indicating that these variables had higher explanatory power than the rest of the variables. The PDSI, a surrogate for antecedent conditions, and rainfall totals exhibited relatively weak influence on crop yield. Based on these observations, we chose the top four key variables with a high probability of inclusion (>0.9) to build model 1 and model 2 to simulate rice yield (Table 2).

Our long-term, hierarchical Bayesian model 1 had a DIC of 393 and a classical $R^2$ of 0.82. Fig 5 summarizes the medians and related confidence intervals of regression coefficients of the four key variables (air temperature, area planted, groundwater level, number of irrigation wells) that were used in model 1. S2 Table highlights the descriptive statistics for the model intercept ($\alpha$) and precision ($\tau$). The precision ($\tau$) highlighted the potential uncertainty in crop yield across counties. The confidence intervals of regression coefficients demonstrated that the uncertainty in relationships between rice yield and the key drivers (Fig 5). The median regression coefficient was maximum for the number of irrigation wells, followed by groundwater level, area planted, and air temperature. The number of irrigation wells had a stronger influence than the groundwater level on predicting rice yield. A post predictive check of the model 1 was performed by estimating Pearson correlation coefficient (r) between predicted ($y_{pred}$)

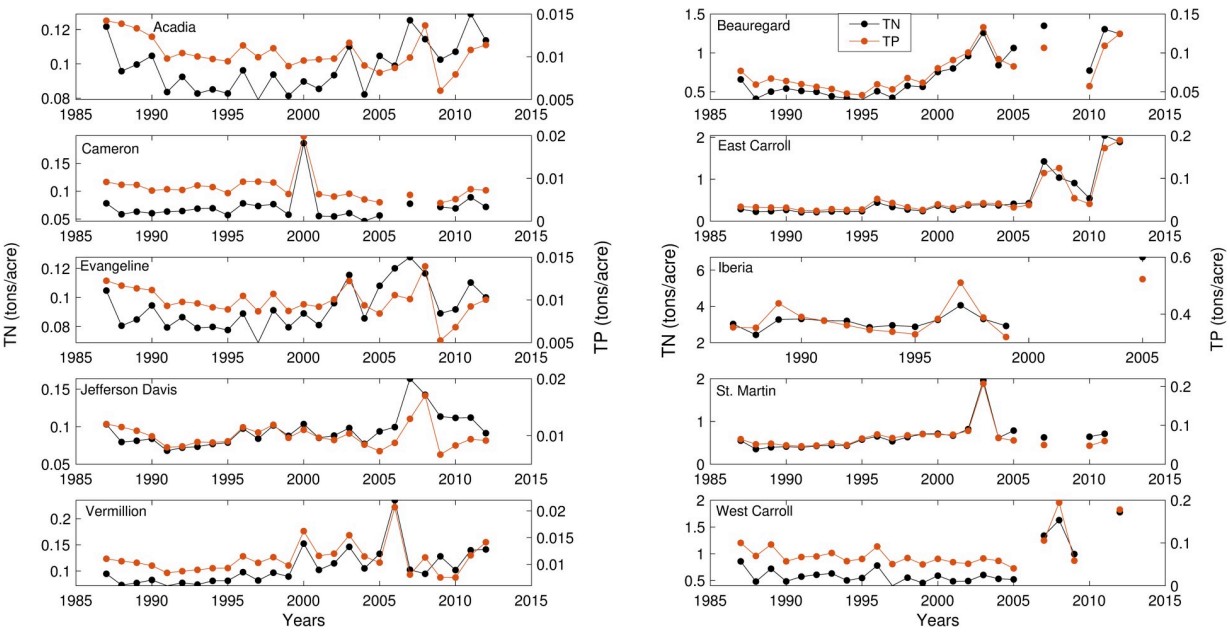

**Fig 4. Nitrogen and Phosphorus fertilizers inputs applied to the study counties.**

and observed (y) was about 0.9, and the 95% confidence interval ranged between 0.88 and 0.92.

Our limited time duration, rice yield model 2 had a DIC of 197 and $R^2$ of 0.64. The distribution of the regression coefficient of N fertilizers was positively related to rice yield but had high uncertainty (S4 Fig). As a post predictive check, model 2 had Pearson correlation coefficient (r) between predicted ($y_{pred}$) and observed (y) of about 0.8, and the 95% confidence interval ranged between 0.74 and 0.84.

## 4. Discussion

Our work is unique in revealing the linkages among food, climate, and groundwater for a region that has been showing increasing rice yield in the past 56 years. The hierarchical Bayesian model 1 successfully simulated rice yield with the selected variables such as groundwater level, area harvested, the number of irrigation wells, and air temperature. The proposed framework was tested for rice, but it could be extended to other crops and other locations.

Crop yield models have shown the negative impacts of antecedent conditions on crop production [66–69]. Our work showed a decline in rice yield during extremely dry conditions

**Table 2. Summary of the probability of inclusion for all variables.**

| Variables | Probability of Inclusion |
|---|---|
| Groundwater level | 1.00 |
| Irrigation Wells | 1.00 |
| Air Temperature | 0.999 |
| Area Planted | 0.999 |
| Oil Price | 0.761 |
| Seasonal Rainfall | 0.265 |
| PDSI | 0.133 |
| Annual Rainfall | 0.061 |

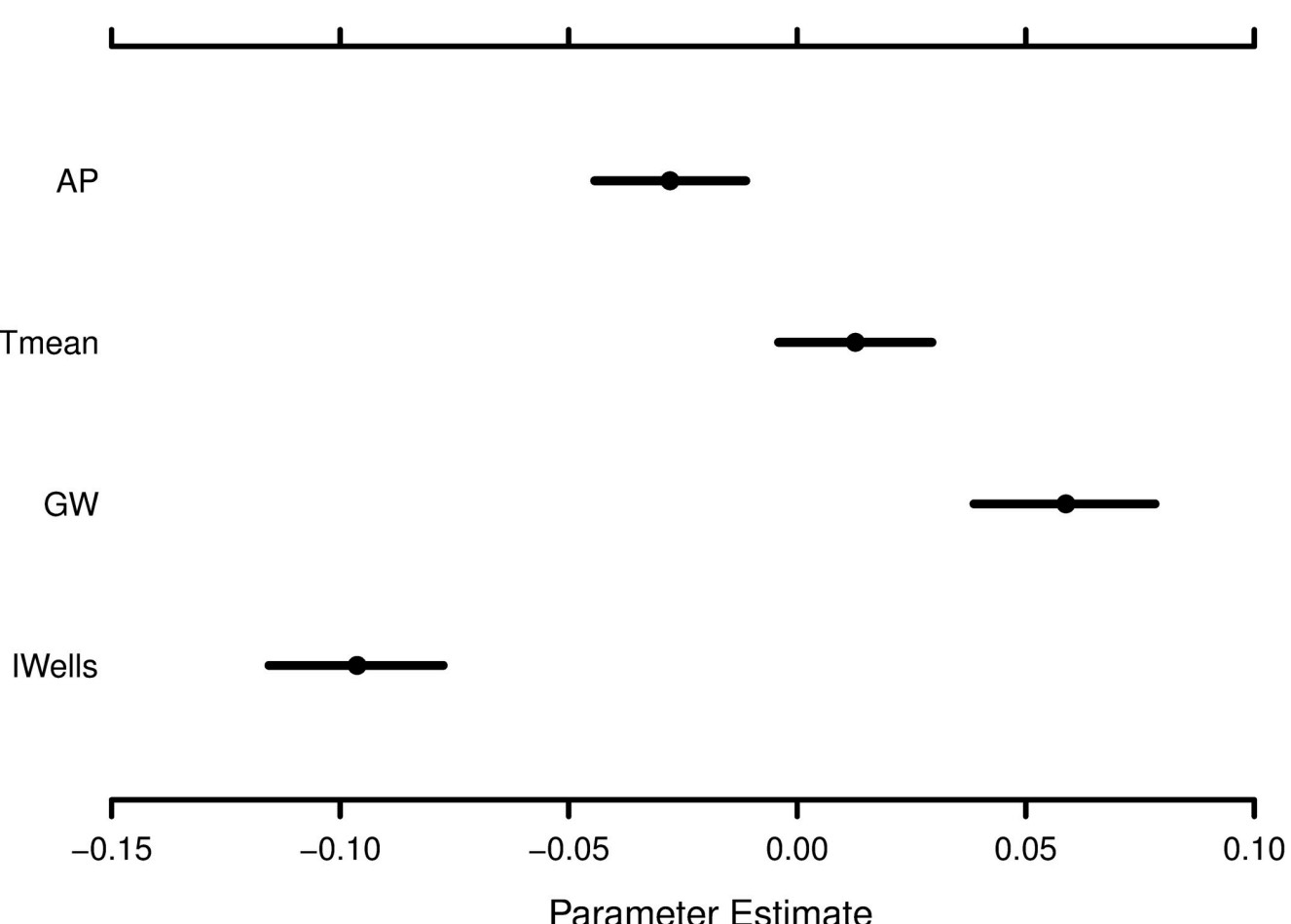

**Fig 5. The posterior distributions of regression coefficients for the covariates used in the hierarchical Bayesian model 1.** Black filled circle and associated thick black line represent median and 50% confidence interval, respectively. Abbreviations: Irrigation wells (Iwells), Area planted (AP), Groundwater level (GW), Mean Air temperature (Tmean).

over a short time scale, but a minimal effect was noted over the long term (Figs 1 and 3). These results are also supported by a global study in which authors demonstrated that the impact of extreme conditions on crop yields is most notable at a short time scale, and the long term patterns of yield are rarely altered [69]. In an attempt to offset the climate-induced demand, farmers increased the installation of irrigation wells, especially during times of frequent dry conditions (Figs 1 and 3). However, the installation of irrigation wells could not buffer the decline in rice yield for the short-term, explaining the negative relationship between rice yield and irrigation wells (Figs 1, 3 and 5). The farmer's ability to install irrigation wells helped them to sustain the yield over the long-term. Our work aligns with a recent study that highlighted the efficacy of irrigation wells in maintaining the economic benefits of crop production for a range of hydro-climatic conditions [70]. These findings underline a need to build short-term and long-term adaption strategies to counter droughts and minimize yield gaps [68].

Utilizing an empirical relationship to simulate crop yield with groundwater has been rare [71]. Groundwater can influence crop growth in multiple ways. For instance, groundwater level may determine the water available for irrigation and plays a critical role in the sustenance of water intensive crops, such as rice. Our findings showed a substantial decline in groundwater level for most study counties (Fig 1), which can be attributed to excessive pumping for

irrigation [35–36]. It is likely that the current rate of groundwater decline may not sustain rice production in the future, highlighting a need to develop sustainable adaption strategies to optimize groundwater usage and to maintain rice yield in the region. An approach may be adopted that is similar to the study where groundwater level and rice yield were linked to propose adaption strategies for an alternate crop and reverse the groundwater declining trends [72]. The empirical relationship between water level and crop yield proposed in our study may help us simulate the impact of plausible changes in groundwater on crop production and prepares us in advance to manage the demand for water. Such empirical relationships are also important from an economic perspective because groundwater is intricately intertwined with global food trade [73].

Area planted or "cropped area" is generally considered an important variable to simulate crop yield [24,69,74]. However, simulating crop yield in response to varying (increasing or decreasing) planted area could be difficult because the production per unit land may depend upon the agricultural management practices and land productivity [11,15,24,65]. Our work showed that cropped area declined or almost remained unchanged (Fig 2), but the rice yield continued to increase over the 56 years (Fig 1). The negative relationship noted between cropped area and rice yield could be attributed to the adaptation of better crop and water management practices by farmers in the region over time [15,75,76]. Farmers in the region have gradually shifted to more productive hybrid cultivars over time [79]. Additionally, there has been a 52% decline in the number of small farms in Louisiana and other rice-dominated regions over the last two decades [65]. The consolidation of farms facilitated the use of advanced precision agricultural equipment, resulting in improved rice yield over time [65]. We suggest that the combination of changes in agricultural management practices and the usage of advanced technologies may have sustained and slightly improved the rice yield, despite the decline in the cropped area. Similar to our work, several studies have reported a negative relationship between crop yield and area planted [24,77,78]. For example, a study attributed increasing rice yield (> 50%) to the use of a more productive cultivar, despite a decline in the cropped area [77]. Overall, disentangling the mechanisms driving the relationship between crop yield and area planted is a multifaceted problem, and highlights a need to incorporate interactions of several agricultural management variables to examine the effect of cropped area on the crop yield.

Our work also showed that the regression coefficients of air temperature could vary widely and revealed the heterogeneity in the relationship between air temperature and rice yield over the 56 years (Fig 5). Air temperature can influence crop yield via multiple pathways, such as by mediating water availability, ecophysiology, and pest infestation [79,80]. Our results are in agreement with studies using process-based crop growth [81,82] and statistical [17,25] models that reported air temperature as an important driver of crop yield by utilizing a range of climate scenarios. For instance, temperature could increase or decrease yield, depending on latitude and crop type [25]. Lastly, our variable importance analysis further confirmed that air temperature was relatively more important variable than rainfall totals (Table 2). This result agrees with a global scale study indicating that the air temperature may have a stronger influence on simulating crop yield than rainfall [25].

The role of fertilizers in augmenting crop growth and increasing yield from regional to global scales has been well documented [83–86]. However, our results showed no unidirectional change in N fertilizer amounts over time (Fig 4). The model 2 with N fertilizer did show a positive relationship with rice yield, but this relationship is subject to high uncertainty (S4 Fig). These findings indicated that the N fertilizer may have a contribution, albeit limited and less important than other variables, to increasing rice yield in this region. The lack of a significant relationship between fertilizer and rice yield could be attributed to limited datasets.

Conversely, there is some evidence from the study region suggesting a shift towards more efficient application of N in rice farms [87]. Therefore, we speculate that due to the high costs associated with fertilizers, farmers are carefully evaluating their fertilizer needs and relying more on other management practices (e.g., hybrid cultivar, technology) to increase yield in this region.

Technological development is an important variable that we were unable to consider directly in the models [65]. We attempted to look into this possible driver by using farm-related income retrieved from NASS as a surrogate. We assumed that the rise in income would allow farmers to afford better equipment, resulting in higher crop productivity. Mean annual farm-related income at the 5yr interval increased by almost two-fold since the inception of the survey in 1997 (S3 Table). Therefore, it is likely that rising farm-related income might have allowed farmers to use more efficient technologies, leading to higher productivity. However, additional data would be needed to include this in the modeling framework.

## 5. Conclusions and implications

The proposed Bayesian-based framework offers a novel way to dynamically model the impact of climate, groundwater, and agricultural-related drivers on food production. The variable selection approach demonstrated that air temperature was a more important climate driver than rainfall totals, indicating the potential sensitivity of rice production to climate change and warmer temperatures in the near future. Oil prices and PDSI had relatively low influence on rice yield. The ability of the farmers to install wells allowed them to buffer the influence of extremely dry conditions on rice yield over the long-term. However, the installation of irrigation wells could not sustain the decline in rice yield in the short-term, which could explain the negative relationship between rice yield and irrigation wells. The rice acreage declined or showed no change, but the rice yield continued to increase, indicating the implementation of efficient crop management practices such as more productive hybrid cultivar and the optimal use of advanced precision agricultural equipment. We did not detect significant influence of N fertilizer on rice yield.

Our findings have implications for food security because rice is grown in approximately 100 countries and fulfills energy requirements for more than 3 billion people worldwide [88]. Understanding the intimate linkages among food-groundwater-climate is critical to framing holistic climate change adaption strategies, especially in the developing world, with limited resources [89]. Another key implication of our work is about the importance of incorporating uncertainty in the relationship between crop yield and associated drivers in the statistical models. Based on the point estimates (i.e., median) and confidence intervals, both rice yield models exhibited some degree of uncertainty in the relationships between yield and covariates. These results pointed to the importance of drawing inferences based on both point and confidence intervals of the posterior distributions.

## Supporting information

**S1 Table. Correlation* matrix for the explanatory variables.**
(DOCX)

**S2 Table. Descriptive statistics for the model parameters.**
(DOCX)

**S3 Table. Farm-related annual mean income for the study counties.**
(DOCX)

**S1 Fig. A map of location of study groundwater wells along with the county and state boundaries for Louisiana.**
(DOCX)

**S2 Fig. Temporal patterns of mean air temperature and rainfall total during growing season for the climate stations located in the southern and northern part of the Louisiana.**
(DOCX)

**S3 Fig. Temporal patterns of oil prices (USD) observed during the study period.**
(DOCX)

**S4 Fig. The posterior distributions of regression coefficients for the covariates used in the limited duration hierarchical Bayesian model 2.** Black filled circle and associated thick black line represent median and 50% confidence interval, respectively. Abbreviations: Irrigation wells (Iwells), Area planted (AP), Groundwater level (GW), Mean Air temperature (Tmean), and area normalized Nitrogen fertilizers (N_fert).
(DOCX)

## Acknowledgments

The datasets used in the study are publicly available from NCEI[44], NASS[46], USGS[49], and Louisiana DNR[48]. We thank the anonymous reviewer (s) and Dr. Gurpal Toor for their valuable feedback.

## Author Contributions

**Conceptualization:** Nitin K. Singh, Ruchi Bhattacharya, David M. Borrok.

**Data curation:** Nitin K. Singh.

**Formal analysis:** Nitin K. Singh, Ruchi Bhattacharya.

**Funding acquisition:** David M. Borrok.

**Methodology:** Nitin K. Singh, Ruchi Bhattacharya.

**Visualization:** Nitin K. Singh.

**Writing – original draft:** Nitin K. Singh.

**Writing – review & editing:** Nitin K. Singh, Ruchi Bhattacharya, David M. Borrok.

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
