## [Decision Letter · Decision Letter 0]

11 Jun 2020

PONE-D-20-10256

A Bayesian framework to unravel food, groundwater, and climate linkages

PLOS ONE

Dear Dr. Singh,

Thank you for submitting your manuscript to PLOS ONE. After careful consideration, we feel that it has merit but does not fully meet PLOS ONE’s publication criteria as it currently stands. Therefore, we invite you to submit a revised version of the manuscript that addresses the points raised during the review process.

We look forward to receiving your revised manuscript.

Kind regards,

Gurpal S. Toor, Ph.D.

Academic Editor

PLOS ONE

Journal Requirements:

Additional Editor Comments (if provided):

Dear authors,

We received one review report on your manuscript. Unfortunately, another reviewer committed to reviewing the manuscript but did not provide a review. To not significantly delay the decision, I'm forwarding you one reviewer's report who recommended a major revision. Please make sure to address all of the comments. It is likely that this manuscript will be sent for the second round of review.

Best wishes.

Gurpal Toor

Reviewers' comments:

Reviewer's Responses to Questions

**Comments to the Author**

1. Is the manuscript technically sound, and do the data support the conclusions?

Reviewer #1: Partly

2. Has the statistical analysis been performed appropriately and rigorously? 

Reviewer #1: No

3. Have the authors made all data underlying the findings in their manuscript fully available?

Reviewer #1: Yes

4. Is the manuscript presented in an intelligible fashion and written in standard English?

Reviewer #1: Yes

5. Review Comments to the Author

Reviewer #1: This manuscript reports on a statistical regression approach to relate rice yields to groundwater use and other explanatory variables. The research questions are important and some high quality data are used here. However, there are several major methodological questions outstanding that preclude me from recommending the work in its present state. Some suggestions for improvement are included below.

Major comments

1. The literature review of yield models is not convincingly recent. Of the 11 references cited on this topic (refs 7-16), only 2 are from within the past 10 years. This requires a significant update.

2. Similarly, the literature on groundwater and crop yields is far more extensive than is implied here (lines 119-121). There are many papers about the linkages between crop yields and groundwater depletion, for example in the High Plains and North China Plain.

3. There is a significant disconnect regarding the spatial resolution of the study. Most of the paper emphasizes that county level data were used, but then it is just briefly and casually mentioned that because of unspecified data limitations the model is aggregated for the entire study area. Section 2.2 (Datasets) and Table 1 emphasize that county level data were used. But later (lines 213-214) it is admitted that the data were all aggregated. This should be made clear early on because in the end there was not actually any investigation of county-scale effects, which is not the impression given by Table 1. Figures 1 and 2 shows plenty of county level data, so it is not convincing that there is a need to aggregate. It raises questions about why county level models could not be run.

4. The authors are overly focused on “groundwater” when what is actually important here is irrigation to meet crop water demands. Irrigation occurs when there is a rainfall deficit. The way rainfall is used as a variable is relatively naïve. The crop yield is controlled by the total water supplied. When rainfall was low, irrigation was higher. So the real variable should be the sum of rainfall plus irrigation. The only time when using just rainfall would make sense to use a variable is if there was a control site that had no additional irrigation.

5. The statement on line 349 that area planted affects yield per acre does not make sense. Area planted affects groundwater level because this is an integrative quantity. But yield should be normalized to area planted. Therefore more elaboration is required to explain what mechanism is proposed to account for area planted affecting yield. The authors should carefully evaluate how “yield” is defined here because it should be mass per acre planted in that crop.

6. Figure 5 shows the main results - but the values here do not make sense. The coefficients for two important variables are negative, but these should not be negative. These results imply more irrigation wells = lower crop yield? More area planted = lower crop yield? The authors have not explained these at all. Probably because these results are actually not meaningful. The confidence in the overall results is heavily diminished by these values, and further by the authors’ failure to address them.

7. Figures 1 and 2 show some of the explanatory variables over time, but not all. The important variables should all be shown here. Since temperature is thought to be important, this one should be shown as well. Figure S2 shows temperature – and the result is that there is no clear pattern. That is, it is fairly clear that this should have no meaningful impact on the obvious patterns shown in Figure 1. Indeed the results in Figure 5 reveal that the coefficient for air temperature is negligible (the confidence interval for the coefficient includes 0.0). The authors have heavily emphasized the air temperature “finding” in the discussion, however this seems not justified.

8. The work is mostly dependent on the assumption that irrigation from groundwater caused the increase in rice yield. It is indeed a compelling correlation in Figure 1. However, other drivers may well have similar behavior. The most important one to consider as well is fertilizer input. The authors introduce a cursory analysis of fertilizer, but only at the end of the paper. This analysis needs to be included as a major component of the study, not just an add-on at the end. From Figure S4, the fertilizer data is shown as just tons, rather than mass per hectare which is how it should be included in the model. Based on the declining trend of area planted (Figure 2), it seems fertilizer per area has increased. This should definitely show an effect in the model. Even though these data are a shorter time period than the overall dataset, a second model for just this period (from 1987) should be created to more carefully assess the role of fertilizer vs irrigation.

Other comments

Line 14, Before even reading the paper, it is not a good sign when the email address provided does not correspond to the listed affiliation.

Title: The paper is about rice in Louisiana, so the title should reflect this. It can be a detriment when the title includes overly abstract (alternatively, overly grandiose) claims that are actually beyond the scope of what was done.

Lines 59-63, There are mixed messages here about the variable importance. Clarify with a straightforward statement about the relative importance.

Line 93, 60% compared to when?

Table 1, suggest to list reference number for data sources

Line 191, define which months are the growing season

Lines 210 and 230, “We built”

Line 221, The use of “N” implies a normal distribution. But apparently a gamma distribution was used (line 227), so a different symbol should be chosen.

Ref 29, not enough info provided here

Figure 1, Replace “groundwater level” with “depth to groundwater”

Figures 2 and 3 are switched.

Lines 315-316, This says rainfall at county scale, but the actual model was aggregated, not county scale.

Figure 4, I found this to be not useful at all. What is the intended message here? A summary of the mean/std of these variables would suffice.

6. PLOS authors have the option to publish the peer review history of their article (what does this mean?). If published, this will include your full peer review and any attached files.

Reviewer #1: No

---

## [Author Response · Author response to Decision Letter 0]

9 Jul 2020

July 8, 2020

Dr. Gurpal Toor

Editor, PLOS One

San Francisco, CA 

USA

Dear Dr. Toor,

Thank you for facilitating the peer-review process. We thank the anonymous reviewer for the detailed feedback. We have incorporated suggestions made by the reviewer, and believe the paper is much stronger. The tracked changes in the draft highlights the modifications made to the original version.

Specifically, the Introduction section has been expanded to include more works on rice models and the connection between groundwater and crop production. We built a second Bayesian model to explore the effect of fertilizer on rice yield for the limited-duration when fertilizer data were available. The Discussion section has been updated to explain the regression coefficients of area planted and irrigation wells. Lastly, we have clarified the confusion regarding the spatial scale of modeling. We think the manuscript now offers a clearer message, and our results are better situated in the larger context of food, water, and climate nexus.

Sincerely, 

Nitin Singh, Ph.D., 

(On behalf of all co-authors)

Dear authors,

We received one review report on your manuscript. Unfortunately, another reviewer committed to reviewing the manuscript but did not provide a review. To not significantly delay the decision, I'm forwarding you one reviewer's report who recommended a major revision. Please make sure to address all of the comments. It is likely that this manuscript will be sent for the second round of review.

Best wishes.

Gurpal Toor

5. Review Comments to the Author

Reviewer #1: This manuscript reports on a statistical regression approach to relate rice yields to groundwater use and other explanatory variables. The research questions are important and some high quality data are used here. However, there are several major methodological questions outstanding that preclude me from recommending the work in its present state. Some suggestions for improvement are included below.

Response: Thank you for providing the detailed feedback. We appreciate your suggestions in improving the quality of the manuscript. 

Major comments

Comment#1. The literature review of yield models is not convincingly recent. Of the 11 references cited on this topic (refs 7-16), only 2 are from within the past 10 years. This requires a significant update.

Response: 

As suggested, we have added eight recent references to the Introduction section. Please refer to lines 104-109 for the details and see the list of citations below.

i) Yu Y, et al. Changes in rice yields in China since 1980 associated with cultivar improvement, climate and crop management. Field Crops Research. 2012 Sep 20;136:65-75.

ii). Ma G, et al. Assimilation of MODIS-LAI into the WOFOST model for forecasting regional winter wheat yield. Mathematical and Computer Modelling. 2013 Aug 1;58(3-4):634-43.

iii) Xiong W, et al. Can climate-smart agriculture reverse the recent slowing of rice yield growth in China?. Agriculture, ecosystems & environment. 2014 Oct 15;196:125-36. 

iv) Vanuytrecht E, et al. AquaCrop: FAO's crop water productivity and yield response model. Environmental Modelling & Software. 2014 Dec 1;62:351-60.

v) Yang X, et al. Potential benefits of climate change for crop productivity in China. Agricultural and Forest Meteorology. 2015 Aug 15;208:76-84. 

vi). Li T, et al. Uncertainties in predicting rice yield by current crop models under a wide range of climatic conditions. Global change biology. 2015 Mar;21(3):1328-41.

vii). Espe MB, et al. Yield gap analysis of US rice production systems shows opportunities for improvement. Field Crops Research. 2016 Sep 1;196:276-83.

viii) Liu L, et al. Linking field survey with crop modeling to forecast maize yield in smallholder farmers’ fields in Tanzania. Food Security. 2020 Mar 5:1-2.

Comment#2. Similarly, the literature on groundwater and crop yields is far more extensive than is implied here (lines 119-121). There are many papers about the linkages between crop yields and groundwater depletion, for example in the High Plains and North China Plain. 

Response: We have updated the literature with more work from US high plains and North China Plains. Please refer to lines 124-128 for details and see the list of citations below.

i) Sun Q, et al. Optimization of yield and water-use of different cropping systems for sustainable groundwater use in North China Plain. Agricultural Water Management. 2011 Mar 1;98(5):808-14.

ii) Steward DR, et al. Tapping unsustainable groundwater stores for agricultural production in the High Plains Aquifer of Kansas, projections to 2110. Proceedings of the National Academy of Sciences. 2013 Sep 10;110(37):E3477-86.

iii) Pei H et al. Impacts of varying agricultural intensification on crop yield and groundwater resources: comparison of the North China Plain and US High Plains. Environmental Research Letters. 2015 Apr 20;10(4):044013.

iv) Jägermeyr J, et al. Integrated crop water management might sustainably halve the global food gap. Environmental Research Letters. 2016 Feb 16;11(2):025002.

v) van Oort PAJ. et al. Towards groundwater neutral cropping systems in the Alluvial 597 Fans of the North China Plain. Agricultural Water Management 165: 131-140.

vi) Zhong H, et al. Mission Impossible? Maintaining regional grain production level and recovering local groundwater table by cropping system adaptation across the North China Plain. Agricultural Water Management. 2017 Nov 1;193:1-2. 

vii) Cotterman KA, et al. Groundwater depletion and climate change: future prospects of crop production in the Central High Plains Aquifer. Climatic change. 2018 Jan 1;146(1-2):187-200.

Comment#3. There is a significant disconnect regarding the spatial resolution of the study. Most of the paper emphasizes that county level data were used, but then it is just briefly and casually mentioned that because of unspecified data limitations the model is aggregated for the entire study area. Section 2.2 (Datasets) and Table 1 emphasize that county level data were used. But later (lines 213-214) it is admitted that the data were all aggregated. This should be made clear early on because in the end there was not actually any investigation of county-scale effects, which is not the impression given by Table 1. Figures 1 and 2 shows plenty of county level data, so it is not convincing that there is a need to aggregate. It raises questions about why county level models could not be run. [County level model?] 

Response: Thanks for pointing this out. The draft has been updated throughout to address the confusion regarding the spatial scale of modeling. As suggested, the spatial scale of work has also been added in the Introduction (line 150) and Method (lines 222-224) sections. County-level Bayesian models could not be developed due to missing datasets in groundwater and agricultural-related variables (e.g., number of irrigation wells). The missing data resulted in as few as ~ 20 observations per county when all dependent and response variables were available. Thus, we aggregated all county level observations to build a regional scale Bayesian model. Due to similar climate and agricultural management practices among adjacent counties, we do not expect to see spatially variable relationships at the county level between the major drivers and rice yield. In essence, the total region of Louisiana rice production is relatively small with relatively similar properties, such that breaking it down further (if it were possible) is not expected to add much. 

Comment#4. The authors are overly focused on “groundwater” when what is actually important here is irrigation to meet crop water demands. Irrigation occurs when there is a rainfall deficit. The way rainfall is used as a variable is relatively naïve. The crop yield is controlled by the total water supplied. When rainfall was low, irrigation was higher. So the real variable should be the sum of rainfall plus irrigation. The only time when using just rainfall would make sense to use a variable is if there was a control site that had no additional irrigation. 

Response: Unfortunately, irrigation datasets for the study region are only available through USGS at 5-year intervals from 1985 to 2015 for the study counties, resulting in insufficient observations (<=7) for the modeling. Farmers in the state mostly rely on irrigation to meet their rice water demands and more than 90% of irrigation is sustained via groundwater withdrawal in this region [Sargent, 2011; Vories and Evett, 2014]. This widespread overdrafting has led to a decline in groundwater up-to 40 feet in the state [Reilly et al., 2010]. This means that groundwater levels are generally correlated with irrigation (seasonally and annually). Thus, based on the prior research and groundwater-dominated irrigation practices in the region, we hypothesized that groundwater level can serve as a reasonable substitute to irrigation. This hypothesis is supported by our recent data-intensive modeling work demonstrated ‘causal’ linkages between groundwater level and rice yield, indicating that groundwater can influence rice yield over long-term in Louisiana [Singh and Borrok, 2019]. It is true that rainfall does influence the need for irrigation, but even in wet years, there is substantial groundwater use for irrigation for rice in Louisiana. For these reasons, we think our choices of variables for the model are reasonable. 

Comment#5. The statement on line 349 that area planted affects yield per acre does not make sense. Area planted affects groundwater level because this is an integrative quantity. But yield should be normalized to area planted. Therefore more elaboration is required to explain what mechanism is proposed to account for area planted affecting yield. The authors should carefully evaluate how “yield” is defined here because it should be mass per acre planted in that crop.

Response: We want to clarify that crop yield is a standard crop commodity metric reported by the USDA, and it is normalized by the area planted. We have included more text in the Discussion section to expand on how area planted and crop yield are linked. Please refer to lines 375-393. 

Comment#6. Figure 5 shows the main results - but the values here do not make sense. The coefficients for two important variables are negative, but these should not be negative. These results imply more irrigation wells = lower crop yield? More area planted = lower crop yield? The authors have not explained these at all. Probably because these results are actually not meaningful. The confidence in the overall results is heavily diminished by these values, and further by the authors’ failure to address them.

Response: We apologize for the confusion. We have added text in the Discussion section to explain the regression coefficients linking irrigation wells and area planted with rice yield. We have also cited other researchers who have found a similar relationship elsewhere. Please refer to lines 345-358 and lines 375-393 for details. 

Briefly, the negative correlation of area planted and lower yield could be attributed to increased technological efficiency and better cultivars over time that sustained and slightly improved the rice yield, despite the decline in the cropped area [Nalley et al., 2016; Espe et al., 2016; McBride et al., 2018]. One reason why irrigation wells negatively correlated with crop yield may be that more wells were drilled in drought years when crop yield was lower so they would be available to combat future drought conditions.

Comment#7. Figures 1 and 2 show some of the explanatory variables over time, but not all. The important variables should all be shown here. Since temperature is thought to be important, this one should be shown as well. Figure S2 shows temperature – and the result is that there is no clear pattern. That is, it is fairly clear that this should have no meaningful impact on the obvious patterns shown in Figure 1. Indeed the results in Figure 5 reveal that the coefficient for air temperature is negligible (the confidence interval for the coefficient includes 0.0). The authors have heavily emphasized the air temperature “finding” in the discussion, however this seems not justified.

Response: We agree that there is relatively high uncertainty in relationship between rice yield and temperature. Per suggestion, we have reduced the text regarding temperature’s influence on rice yield. Please see the updated text at lines 394-404. 

Comment# 8. The work is mostly dependent on the assumption that irrigation from groundwater caused the increase in rice yield. It is indeed a compelling correlation in Figure 1. However, other drivers may well have similar behavior. The most important one to consider as well is fertilizer input. The authors introduce a cursory analysis of fertilizer, but only at the end of the paper. This analysis needs to be included as a major component of the study, not just an add-on at the end. From Figure S4, the fertilizer data is shown as just tons, rather than mass per hectare which is how it should be included in the model. Based on the declining trend of area planted (Figure 2), it seems fertilizer per area has increased. This should definitely show an effect in the model. Even though these data are a shorter time period than the overall dataset, a second model for just this period (from 1987) should be created to more carefully assess the role of fertilizer vs irrigation.

Response: The patterns of area normalized N/P fertilizer did not show consistent, unidirectional patterns across counties (Figure 4). However, we did develop a second, limited duration, Bayesian model that includes data for N fertilizer used in addition to the selected key drivers when fertilizer datasets were available. Briefly, the regression coefficient of fertilizer input was positively related to rice yield but had high uncertainty (Figure S4). These findings indicated that the N fertilizer might have a limited contribution to increasing rice yield in this region. Please refer to the Methods (lines 205-207; 235-241), Results (lines 264-268; 311-315) and Discussion section (lines 405-427) for details.

Comment#9 Line 14, Before even reading the paper, it is not a good sign when the email address provided does not correspond to the listed affiliation.

Response: Done 

Comment#10 Title: The paper is about rice in Louisiana, so the title should reflect this. It can be a detriment when the title includes overly abstract (alternatively, overly grandiose) claims that are actually beyond the scope of what was done.

Response: We have modified the title. 

Comment#11 Lines 59-63, There are mixed messages here about the variable importance. Clarify with a straightforward statement about the relative importance.

Response: Done

Comment#12 Line 93, 60% compared to when?

Response: We have clarified the year in the statement.

Comment#13 Table 1, suggest to list reference number for data sources

Response: Done

Comment#14 Line 191, define which months are the growing season

Response: Please refer to lines 204-205.

“The growing season for the rice in this region is February through July.” 

Comment#15 Lines 210 and 230, “We built”

Response: Fixed

Comment#16 Line 221, The use of “N” implies a normal distribution. But apparently a gamma distribution was used (line 227), so a different symbol should be chosen.

Response: We want to clarify that the normal distribution was used for most of the parameters in data and process models. The gamma distribution was only used for the precision (i.e., tau). Please refer to lines 240-241. 

Comment#17 Ref 29, not enough info provided here

Response: We have added more details regarding aquifer systems. Please refer to lines 186-193.

Comment#18 Figure 1, Replace “groundwater level” with “depth to groundwater”

Response: Done

Comment#19 Figures 2 and 3 are switched.

Response: Thanks for catching this. Done

Comment#20

Lines 315-316, This says rainfall at county scale, but the actual model was aggregated, not county scale.

Response: We have addressed the confusion regarding the spatial scale of modeling throughout the draft. 

Comment#21

Figure 4, I found this to be not useful at all. What is the intended message here? A summary of the mean/std of these variables would suffice.

Response: The descriptive statistics of the parameters have been included in the supplementals (Table S2).

References:

Espe MB, Cassman KG, Yang H, Guilpart N, Grassini P, Van Wart J, Anders M, Beighley D, Harrell D, Linscombe S, McKenzie K. Yield gap analysis of US rice production systems shows opportunities for improvement. Field Crops Research. 2016 Sep 1;196:276-83.

McBride W, Skorbiansky SR, Childs N. US Rice Production in the New Millennium: Changes in Structure, Practices, and Costs. United States Department of Agriculture, Economic Research Service; 2018 Dec.

Nalley L, Tack J, Barkley A, Jagadish K, Brye K. Quantifying the agronomic and economic performance of hybrid and conventional rice varieties. Agronomy Journal. 2016 Jul;108(4):1514-23.

Reilly T E, Dennehy K F, Alley W M and Cunningham W L 2008 Ground-water availability in the United States (No. 1323) Geological Survey (US)

Sargent, B.P., Revised 2012. Water use in Louisiana, 2010. Louisiana Department of Transportation and Development, Water Resources Special Report 17, p. 145. 2011

Singh NK, Borrok DM. A Granger causality analysis of groundwater patterns over a half-century. Nature Scientific reports. 2019 Sep 6;9(1):1-8.

Vories ED, Evett SR. Irrigation challenges in the sub-humid US Mid-South. International Journal of Water. 2014;8(3):259-74.

---

## [Editor Report · Decision Letter 1]

14 Jul 2020

A Bayesian framework to unravel food, groundwater, and climate linkages: A case study from Louisiana

PONE-D-20-10256R1

Dear Dr. Singh,

We’re pleased to inform you that your manuscript has been judged scientifically suitable for publication and will be formally accepted for publication once it meets all outstanding technical requirements.

Kind regards,

Gurpal S. Toor, Ph.D.

Academic Editor

PLOS ONE

Additional Editor Comments (optional):

Thank you for thoroughly addressing the reviewer's comments and providing justification for comments that were not addressable. The manuscript is a good shape now to be accepted. Congratulations!
---

## [Editor Report · Acceptance letter]

17 Jul 2020

PONE-D-20-10256R1 

A Bayesian framework to unravel food, groundwater, and climate linkages: A case study from Louisiana 

Dear Dr. Singh:

I'm pleased to inform you that your manuscript has been deemed suitable for publication in PLOS ONE. Congratulations! Your manuscript is now with our production department. 

Kind regards, 

on behalf of

Dr. Gurpal S. Toor 

Academic Editor

PLOS ONE